# COVID-19 and the kidney: A retrospective analysis of 37 critically ill patients using machine learning

**Anna Laura Herzog**[1]*, **Holger K. von Jouanne-Diedrich**[2]*, **Christoph Wanner**[3], **Dirk Weismann**[4], **Tobias Schlesinger**[5], **Patrick Meybohm**[5], **Jan Stumpner**[5]

**1** Division of Nephrology, Medizinische Klinik I, Transplantationszentrum, University of Würzburg, University Hospital Wuerzburg, Würzburg, Germany, **2** Faculty of Engineering, Competence Centre for Artificial Intelligence, TH Aschaffenburg (University of Applied Sciences), Aschaffenburg, Germany, **3** Division of Nephrology, Medizinische Klinik I, University of Würzburg, University Hospital Wuerzburg, Würzburg, Germany, **4** Intensive Care Unit, Medizinische Klinik I, University of Würzburg, University Hospital Wuerzburg, Würzburg, Germany, **5** Department of Anaesthesiology and Intensive Care, University of Würzburg, University Hospital Wuerzburg, Würzburg, Germany

* Herzog_A1@ukw.de (ALH); Holgervon.Jouanne-Diedrich@th-ab.de (HKVJD)

## Abstract

### Introduction

There is evidence that SARS-CoV2 has a particular affinity for kidney tissue and is often associated with kidney failure.

### Methods

We assessed whether proteinuria can be predictive of kidney failure, the development of chronic kidney disease, and mortality in 37 critically ill COVID-19 patients. We used machine learning (ML) methods as decision trees and cut-off points created by the OneR package to add new aspects, even in smaller cohorts.

### Results

Among a total of 37 patients, 24 suffered higher-grade renal failure, 20 of whom required kidney replacement therapy. More than 40% of patients remained on hemodialysis after intensive care unit discharge or died (27%). Due to frequent anuria proteinuria measured in two-thirds of the patients, it was not predictive for the investigated endpoints; albuminuria was higher in patients with AKI 3, but the difference was not significant. ML found cut-off points of >31.4 kg/$m^2$ for BMI and >69 years for age, constructed decision trees with great accuracy, and identified highly predictive variables for outcome and remaining chronic kidney disease.

### Conclusions

Different ML methods and their clinical application, especially decision trees, can provide valuable support for clinical decisions. Presence of proteinuria was not predictive of CKD or AKI and should be confirmed in a larger cohort.

**Data Availability Statement:** The raw data from this study are available upon request. In our study the need for informed consent from individual

patients was waived by the ethics committee of the university of Wuerzburg. Data contain potentially identifying and sensitive patient information. According to local requirements, anonymized individual patient level data can only be shared upon request and prior written permission from the Ethics Committee of the University of Wuerzburg (Versbacher Str. 9, 97078 Wuerzburg, Germany, ethikkommission@uniwuerzburg.de) and the data protection officer of the University hospital of Wuerzburg. All other data are provided in the manuscript. The entire R-Package can be downloaded under https://github.com/vonjd/OneR, the link is of course provided.

**Funding:** The authors received no specific funding for this work.

**Competing interests:** Christoph Wanner received honoraria for steering committee membership and lecturing outside the present work from AstraZeneca, Bayer, Boehringer-Ingelheim, Eli Lilly, Mundipharma, and MSD. This does not alter our adherence to PLOS ONE policies on sharing data and materials. All other authors have nothing to declare.

## Introduction

In late 2019, a new type of lung disease caused by a previously unknown coronavirus, SARS-CoV2, appeared for the first time in Wuhan, China. This has now become a global pandemic, infecting more than 50 million people worldwide and causing more than 1.2 million deaths by November 2020. In Germany, a highly industrialized European nation with 82 million inhabitants, more than 700,000 infections and almost 12,000 deaths have been reported until the end of October [1].

Eighty percent of those infected suffer from mild symptoms, such as dyspnea (21.9%), coughing (68.6%), fever (88.5%), myalgia (35.8%), and anosmia (47%) [2–4]. Clinical worsening may occur in 20%, often from 7–10 days after the onset of the disease. Approximately 5% of affected patients require intensive care, with mortality varying between 3% and 50% depending on local factors [5–8].

Older age, underlying hypertension, high cytokine levels (interleukin [IL]-2R, IL-6, IL-10, and tumor necrosis factor [TNF]-$\alpha$), and high ferritin levels are significantly associated with severe coronavirus disease 2019 (COVID-19). The estimated mortality is 1.1% in non-severe patients and 32.5% in severe cases during an average 32 days of follow-up [9]. Underlying cardiac or cerebrovascular disease and elevated cardiac troponin also seem to be as predictive [10,11] as hypertension [5], male sex, cardiac injury, and hyperglycemia for severe COVID-19 [9].

The incidence of acute kidney injury (AKI) is high in critically ill patients, affecting almost 60% [12], and seems to be common among severe COVID-19 cases, affecting approximately 20–40% of patients admitted to intensive care [13].

We assessed 37 critically ill patients treated in University Hospital Wuerzburg to evaluate whether proteinuria or albuminuria can be predictive for developing AKI, CKD, or higher mortality. In addition to classical statistical analysis, we used machine learning (ML) methods. To the best of our knowledge, this adds a new dimension to the body of COVID-19 research.

ML is the study of computer algorithms that allow computer programs to automatically improve through experience [14]. The subarea supervised learning is of concern to us here. Supervised learning is the ML task of learning a function that maps an input to an output based on example input-output pairs [15]. The tasks at hand are more specifically classification problems, with the aim to learn the boundary separating the instances of one class from the instances of other classes [16].

We also used the One Rule classification algorithm [17] implemented in the OneR package [18] and Classification and Regression Trees (CARTs) [19] implemented in the rpart package [20].

## Patients and methods

Thirty-seven critically ill patients were treated between March and May 2020 in the intensive care unit (ICU) of the Department of Anesthesia and Critical Care and the Department of Internal Medicine I, University Hospital Wuerzburg, Wuerzburg, Germany. The medium patient age was 63 years (range 36 to 84 years) and 76% were male. Time of invasive ventilation was 19 days on average (range 4 to 63 days), and two patients had no invasive ventilation. Patient demographics are given in Table 1. The institutional ethics board of the University of Würzburg approved the study. The need for informed consent from individual patients was waived.

We assessed the sex, age, body mass index (BMI), selective (albuminuria) and non-selective proteinuria (2/3 of patients), troponin T and invasive treatment, such as continuous venovenous hemodialysis (CVVHD) or kidney replacement therapy (KRT), extracorporeal

**Table 1. Patients demographics.**

| sex | m | 28 | 75.7% |
|---|---|---|---|
| | f | 9 | 24.3% |
| age | y | 63 ± 12 | 36–84 |
| BMI | kg/qm | 29 ± 5.4 | |
| invasive ventilation | d | 19 ± 16.8 | 4–63 |
| weaning | n | 6 | 16.2% |
| death | n | 10 | 27.0% |
| AKI 1 | n | 4 | 10.8% |
| AKI 2 | n | 4 | 10.8% |
| AKI 3 | n | 20 | 54.1% |
| none | n | 9 | 24.3% |
| CVVHDF | n | 22 | 59.5% |
| | d | 17 ± 15.5 | |
| ECMO | n | 12 | 32.4% |
| | d | 12 ± 6.5 | |
| death (1) | n | 9 | 24.3% |
| crit. Ill. (2) | | 17 | 45.9% |
| recovered (3) | | 10 | 27.0% |
| CKD outcome | | | |
| restitution (1) | n | 15 | 40.5% |
| CKD 2–5 (2) | | 6 | 16.2% |
| CKD 5d (3) | | 16 | 43.2% |
| Prot. Sel mg/gCrea | | | |
| <30 (1) | n | 7 | 18.9% |
| 30–300 (2) | | 10 | 27.0% |
| >300 (3) | | 8 | 21.6% |
| n.d. | | 13 | 35.1% |
| Prot. Non.s. mg/gCrea | | | |
| <30 (1) | n | 4 | 10.8% |
| 30–300 (2) | | 6 | 16.2% |
| >300 (3) | | 14 | 37.8% |
| n.d. | | 13 | 35.1% |

AKI, acute kidney injury; BMI, body mass index; CKD, chronic kidney disease; crit. Ill, critical ill; CVVHDF, continuous veno-venous hemodiafiltration; ECMO, extracorporeal membrane oxygenation; n. d., not determined; Prot. Non.s., non-selective proteinuria i.e. albuminuria; Prot. Sel, selective proteinuria.

membrane oxygenation (ECMO), invasive ventilation, state of AKI defined by the Kidney Disease: Improving Global Outcomes (KDIGO) Guidelines, remaining CKD, and NTproBNP. We assessed the clinical condition after intensive care treatment, either death, recovery, or remaining critical illness. Critical illness was defined as requiring transfer to a weaning unit, smaller hospital, or other care facilities, as no patients were initially admitted from nursery homes. CKD at the time of discharge from the ICU was classified as either none or restoration to baseline kidney function (class 1), remaining or worsened CKD without renal replacement therapy (KRT) (class 2), or prolonged need for KRT after ICU release (class 3). Proteinuria, either selective as albuminuria or non-selective, was defined as none (<30 mg/gCrea), moderately increased (30–300 mg/gCrea), or severely increased (>300 mg/gCrea) according to the KDIGO definition [21] Laboratory findings included troponin T und NTproBNP, which were

collected daily during the first 7 days, on days 10 and 14, and by the time of demission. The highest value was noted.

## Statistical analysis

Statistical analyses were performed using R 4.0.2 [22]. The frequencies of metric variables were expressed as arithmetic mean and standard deviation. If two means of normally distributed data were compared, a two-sided unpaired student's t-test was used. Means from more than two groups were evaluated using analysis of variance (ANOVA) with post-hoc testing (Tukey's test) if significant differences occurred.

In addition to classical statistical analysis, we also used ML methods, namely One Rule (OneR) and decision trees. OneR is a simple classification algorithm that generates one rule for each predictor in the data, and then selects the rule with the smallest total error as its one rule. According to Holte, very simple rules can be expected to perform well for most datasets. An additional advantage is the good interpretability of the resulting rules, which is particularly important in a clinical setting. The OneR package contains improvements over the original OneR algorithm in the form of sophisticated handling of numeric data, which allows for the detection of cut-off values [18]. The OneR algorithm has been previously used successfully in medical research [18,23].

To evaluate the quality of classification algorithms, we used confusion matrices. A confusion matrix is a specific table of statistical classification that maps the performance of an ML algorithm and determines the accuracy of a classification by summing correctly predicted patients over the whole population. In addition, the number of false positives and false negatives can be determined.

## Results

During the COVID-19 pandemic in 2020, we analyzed 37 patients who were treated at our ICUs between March and May. A total of 28 patients suffered acute kidney failure, 20 of them AKI 3 (54.1%); 9 patients died (24.3%), all with AKI 3. The average invasive ventilation time was 19 days (range 4–63 days), and 12 patients (32.4%) were dependent on ECMO for an average of 12 days. Twenty-two patients had to receive KRT (59.5%) for an average of 17 days. Sixteen patients remained on hemodialysis beyond the inpatient stay (43.2%); we did not perform a structured follow-up for more than 60 days. Seventeen patients were transferred to rehabilitation or weaning centers while still critically ill (45.9%; Table 1).

We used the standard t-test to examine the dependence of mortality on COVID-19 and age and found a significantly higher age among deceased patients, as suggested by prior data. Here, the mean value was almost 70 years (range 52 to 84 years), whereas the 27 surviving patients had a mean age of 60 years (range 36 to 80 years; p = 0.05; Fig 1).

A higher BMI also appeared to be associated with increased mortality, but this effect was not significant in our population (p = 0.12). The mean BMI in the survivor group was 28.6 kg/m$^2$ (range 21.5 to 34.7 kg/m$^2$), whereas the mean BMI among deceased patients was 33.4 kg/m$^2$ (range 25.4 to 54.1 kg/m$^2$; Fig 1b).

The optbin function of the OneR package discretizes all numerical data into categorical bins where the cut points are optimally aligned with the target categories. When building a OneR model, this could result in fewer rules with enhanced accuracy. The cutpoints are calculated by pairwise logistic regressions (method "logreg") or as the means of the expected values of the respective classes ("naive"). The function is likely to give unsatisfactory results when the distributions of the respective classes are not (linearly) separable [18].

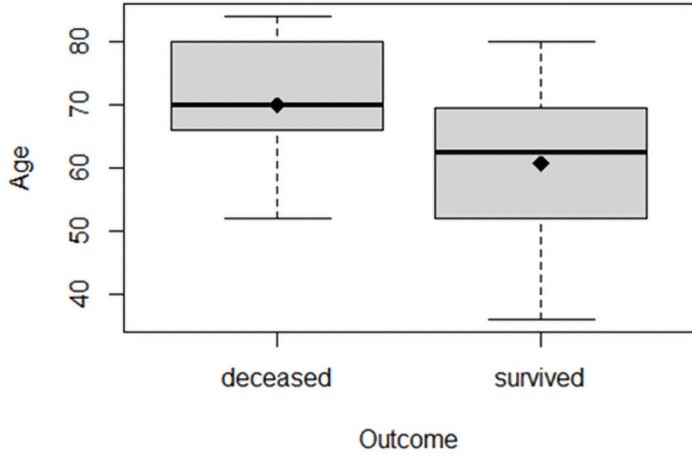

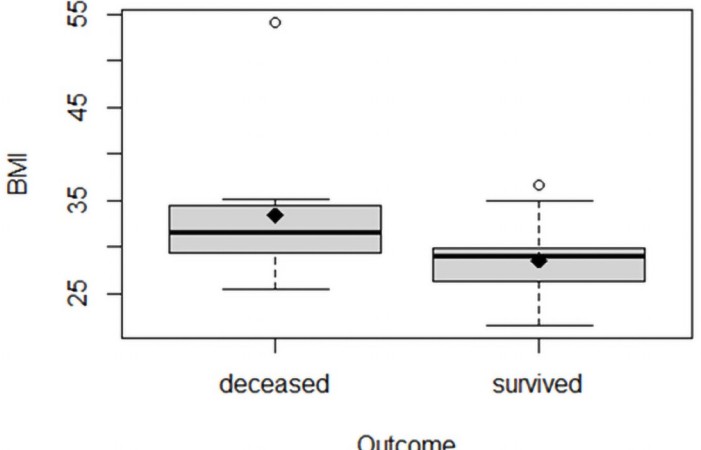

**Fig 1.** a, b: Association between age and BMI and COVID-19 mortality. Data are presented as arithmetic mean and media. The mean in survivors was 60 years, and the mean in deceased was almost 70 years. b, Association between BMI and COVID-19 mortality (p = 0.13). The mean in survivors was 28.5 kg/m2, and the mean in deceased was 33.4 kg/qm, which was not significant.

To obtain an exact cut-off point, we used the OneR algorithm with the optbin function on the same variables. The found rules showed that when the BMI was in the range from 21.6 to 31.4 the patient survived, whereas when the BMI was bigger than 31.4 up to 54.1 the patient died. Those rules have an accuracy of 81.08%.

The resulting diagnostic plot can be seen in Fig 2; the found cut-off point of 31.4 is in line with previous reports [24].

We also conducted urinary analyses for proteinuria. The selective proteinuria (albuminuria) was higher in patients with AKI 3 than in patients with a lower AKI class, but the difference was not significant. However, only nine patients did not suffer acute kidney failure, and three of these patients had no albuminuria test. Overall, albuminuria was slightly lower in this group than in the AKI 1–3 group, but the range of variation was high (Fig 3). In AKI 1, we

## OneR model diagnostic plot

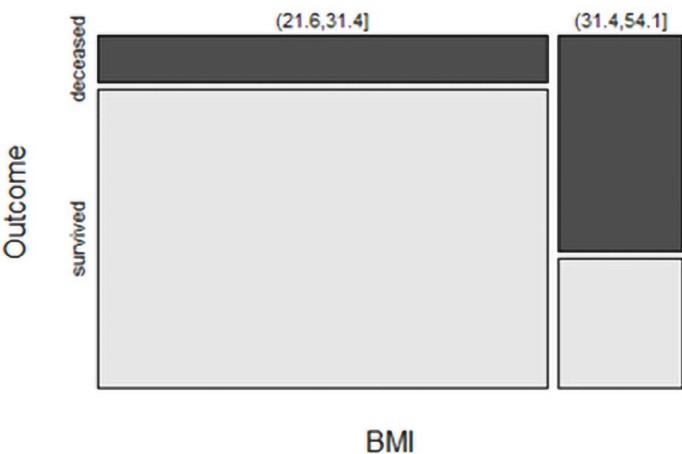

**Fig 2. OneR model diagnostic plot.** Cut-off point for likely death at BMI = 31.4.

found an albuminuria range from 0 to 106 mg/gCrea, in AKI 2 the range was 29 to 112 mg/gCrea, in AKI 3 17 to 500 mg/gCrea, and in the group without AKI the range was 0 to 1660 mg/gCrea.

In the ANOVA regarding proteinuria and albuminuria and the relationship to developing AKI, remaining CKD, or death, we found no significant relationship. The development of AKI was also not significantly related to death or further remaining CKD (Table 2).

The distribution of patients must be taken into account. In some constellations, the case numbers in the individual groups were very small, and proteinuria was not determined for 12 of 37 patients, mostly due to early anuria. As mentioned above, we also constructed various decision trees using COVID-19, AKI, CKD, outcome, NTproBNP, troponin T, BMI, age, and other variables. Fig 4 shows the first binary decision tree with two decision levels; the target variable was the outcome. Based on the presented data, the tree shows that death is very likely

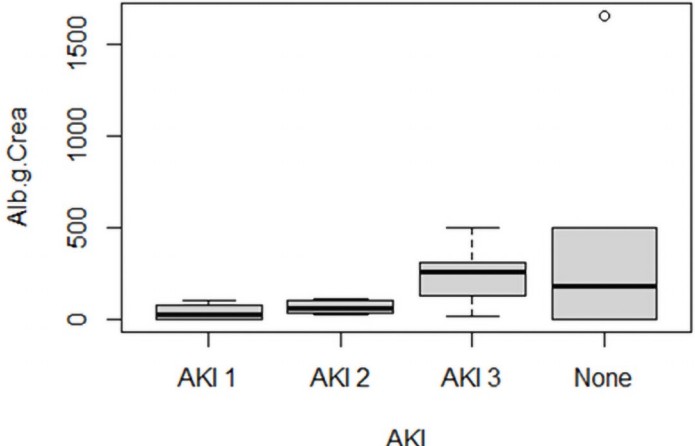

**Fig 3. Selective proteinuria in COVID-19 patients with AKI.** AKI acute kidney injury, 1: 0 to 106 mg/gCrea, AKI 2: 29 to 112 mg/gCrea, AKI 3: 17 to 500 mg/gCrea, None: 0 to 1660 mg/gCrea.

**Table 2. Patients in the individual classification groups of albuminuria and non-selective proteinuria regarding AKI, outcome and CKD.**

| Albuminuria (class) | 1 (n) | 2 (n) | 3 (n) | n.d. | P value |
|---|---|---|---|---|---|
| **AKI 1** | 2 | 2 | 0 | 0 | |
| **AKI 2** | 2 | 2 | 0 | 0 | |
| **AKI 3** | 1 | 5 | 5 | 9 | |
| **none** | 2 | 1 | 3 | 3 | **0.13** |
| **death (n = 9)** | 1 | 1 | 2 | 5 | |
| **crit. Ill (n = 15)** | 2 | 6 | 2 | 5 | |
| **recovered (n = 11)** | 4 | 2 | 3 | 2 | **0.59** |
| **CKD outcome** | | | | | |
| **restitution (n = 16)** | 4 | 5 | 3 | 4 | |
| **CKD 2–4 (n = 6)** | 2 | 2 | 0 | 2 | |
| **CKD 5d (n = 15)** | 1 | 3 | 5 | 6 | **0.51** |
| **selective Proteinuria (class)** | 1 | 2 | 3 | none | |
| **AKI 1** | 2 | 1 | 1 | 0 | |
| **AKI 2** | 0 | 1 | 2 | 1 | |
| **AKI 3** | 0 | 3 | 9 | 8 | |
| **none** | 2 | 1 | 2 | 4 | **0.60** |
| **death (n = 9)** | 0 | 1 | 3 | 5 | |
| **crit. Ill (n = 15)** | 1 | 3 | 6 | 5 | |
| **recovered (n = 11)** | 3 | 2 | 3 | 3 | **0.87** |
| **CKD outcome** | | | | | |
| **restitution (n = 16)** | 3 | 2 | 6 | 5 | |
| **CKD 2–4 (n = 6)** | 1 | 1 | 2 | 2 | |
| **CKD 5d (n = 15)** | 0 | 3 | 6 | 6 | **0.87** |

AKI, acute kidney injury; CKD, chronic kidney disease; crit. Ill, critical ill; n.d., not determined.

to occur in cases of severe AKI in combination with cardiac damage, expressed by a strongly elevated or dynamic troponin T level. In this case the cut-off was 88 mg/dl. In less severe AKI or no AKI, initial albuminuria seems to determine whether the patient fully recovers or if critical illness remains. In this tree, it looks as if even higher proteinuria would lead to a complete recovery.

Using new CKD as a dependent variable, the occurrence of AKI and cardiac involvement also determines the extent to which recovery of renal function or long-term dependence on KRT can be expected. In our cohort, the risk of remaining dependent on KRT was increased, in the case of dialysis-requiring AKI 3 according to our algorithm, from 726 ng/ml NTproBNP (i.e., cardiac impairment due to COVID-19). In the last level of the tree, troponin T > 782 µg/l, the algorithm detected good chances for a complete recovery of renal function despite increased NTproBNP. In our cohort, two patients had at least recovered and did not retain CKD despite cardiac affection of renal failure, which is reflected in the lowest level of the right arm in Fig 5. The algorithm recognizes the most meaningful variable and maps it to the individual levels. Multiple considerations are possible. By defining the depth of the tree in advance, non-relevant variables are disregarded. In this case, the proteinuria (neither selective nor non-selective) seems to not be meaningful enough to be used in the CART.

When we correlated proteinuria and remaining CKD with outcome, the algorithm found a higher mortality in patients with extended dependence on KRT. If renal function fully recovers or remains only slightly impaired, the algorithm also predicts that remaining disease with

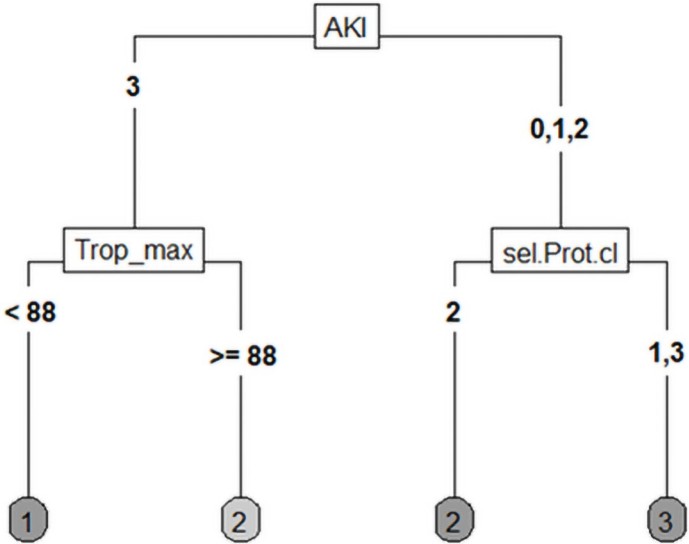

**Fig 4. CART regarding outcome.** Trop_max: Maximum value of troponin T in μg/l, sel.Prot.cI: Selective proteinuria in mg/g creatinine (1: <30 mg/gCrea, 2: 30–300 mg/gCrea, 3: >300 mg/gCrea). Outcome 1: Death, 2: Remaining illness, 3: Recovering.

moderate proteinuria is more common, whereas patients without proteinuria have a better chance of complete recovery. The confusion matrix determined the accuracy with 24 of 37 correctly classified values (64%; Fig 6a and 6b). Prior studies have also found a higher mortality in COVID-19 patients with renal involvement [25,26], but whether proteinuria can be used as a reliable predictive marker is not yet clear [27]. In patients critically ill due to other causes, the presence of proteinuria is certainly a risk factor for the development of AKI and a predictor of higher mortality [28]; for COVID-19 patients, this has yet to be proven in a larger cohort.

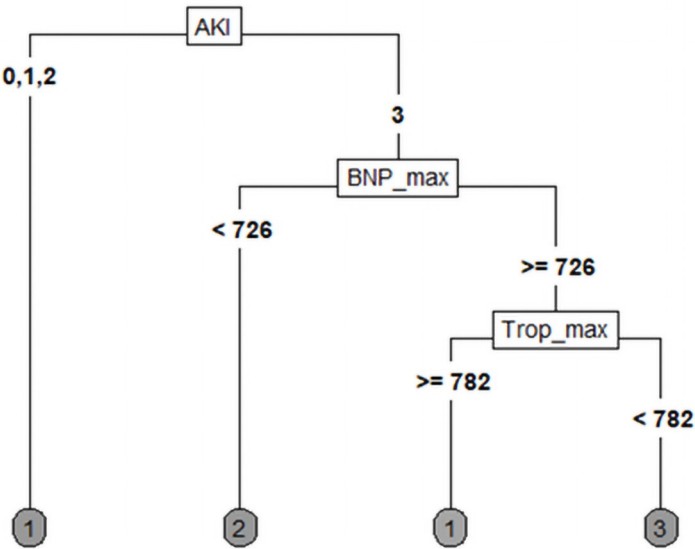

**Fig 5. CKD outcome determined by AKI.** AKI acute kidney injury, maximum NTproBNP value, and maximum troponin T value. 1: Recovery, 2: Remaining or worsened CKD, 3: KRT.

(a) (b)

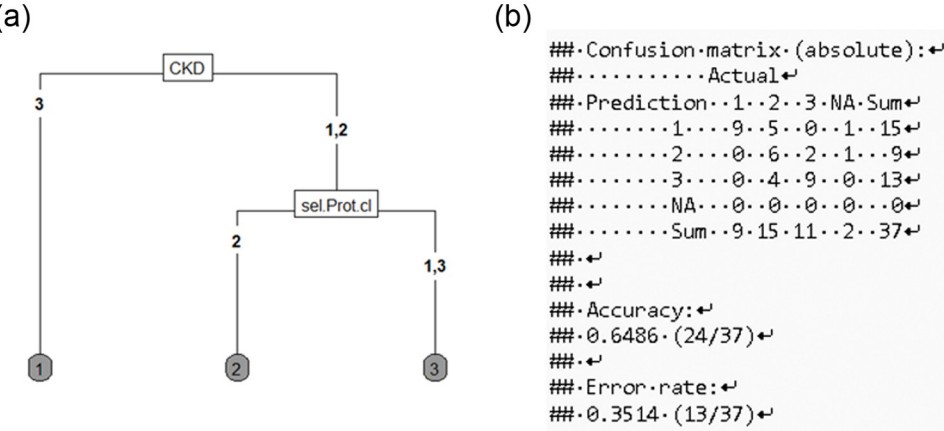

**Fig 6.** a, b: Outcome and its correlation with proteinuria. CKD 1: Restitution to initial function, CKD 2: Remaining new onset or worsening of existing chronic kidney disease, CKD 3: Remaining KRT. Sel. Prot.cl 1: None, 2: 30 mg/gCrea– 300 mg/gCrea, 3: >300 mg/gCrea, Outcome 1: Death, 2: Remaining illness, 3: Recovery. b, Confusion matrix with distribution of the predicted values in absolute numbers. Relative (not shown) describes the percentage deviation.

If we let the algorithm examine the connection between selective or non-selective proteinuria and the expected outcome, the significance of the results seems rather low. The most meaningful variable of the data is shown at the top of the plot, i.e., forms the basis for the division into the first two groups. In the standard ANOVA, the association between proteinuria and mortality in our patient cohort was not significant, but the number of patients and number of proteinuria values were probably too small.

An important consideration is the depth of the tree. With OneR, decision trees can be built with any depth. In our cohort, it was significantly more likely to require KRT for a longer period of time if dialysis-dependent AKI developed (p<0.05). In this case, the tree consists of only one node, which is easy to interpret but reduces the accuracy of the classification, which is only 70%. The few patients who retained CKD were assigned to one of the other two classes in this two-armed tree (Fig 7).

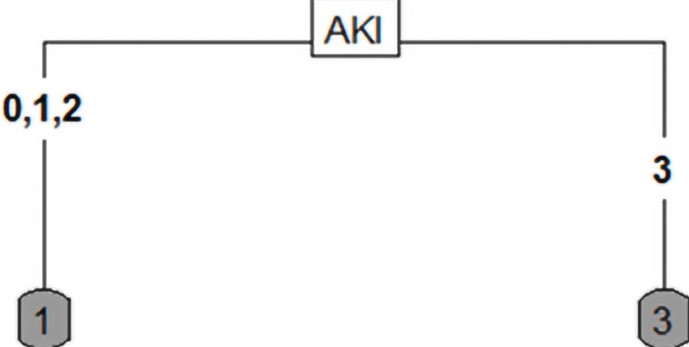

**Fig 7. Remaining CKD, KRT, or recovery depending on the severity of AKI.** CKD 1: Restitution to initial function, CKD 2: New onset or worsening of existing chronic kidney disease, CKD 3: Remaining KRT.

## Discussion

Our observation is that ML methods are adopted reluctantly in medical research, which itself is firmly grounded in classical statistics. Several authors have described the divide between the "two cultures". The main difference in the two approaches can be described by noting that classical statistics assumes that the data are generated by a given stochastic data model, whereas ML uses algorithmic models and treats the data mechanism as unknown [29].

Though classical statistics can be seen as the foundation of all scientific medicine, ML is only used in selected areas, such as intensive care medicine. Some projects facilitate diagnosis, whereas others try to create early warning systems based on a variety of data, increasing the effectiveness of treatment. An ML approach to predicting ICU readmission has been shown to be significantly more accurate than previously published algorithms in internal validation [30]. Many ML systems being used in medical settings are artificial neural networks, mainly in image recognition (i.e., radiology, histology), and are being used to, for example, differentiate between malignant and benign tumors [31,32]. The larger the number of cases, the more accurately ML algorithms can be used, which of course makes the final result more reliable. A limitation of our study is the small number of cases and the fact that urine samples could not be obtained from all patients, especially if anuria has already occurred. Nevertheless, we tried to present a comprehensive overview of different methods and their potential application in clinical routine.

The problem with neural networks is that they constitute so-called black boxes, which means that their decisions cannot be readily explained [33], which constitutes a significant challenge in a medical setting.

Here, we tried to show that both approaches can coexist and complement one another. Interestingly enough, the ML tree-based methods were developed largely by statisticians in the 1970s [34]. We see them as very well equipped to bridge the gap between the "two cultures" because of their firm grounding in classical statistics and convenient availability as mature packages in the R package ecosystem. As we have shown in this paper, tree-based methods are readily comprehensible and can provide new insights, even for data sets that have already been analyzed with more traditional methods.

The general idea of the OneR algorithm is to go through each attribute and evaluate how well it is able to function as a predictor of the dependent variable. The algorithm creates frequency tables for each attribute, providing the number of occurrences at all different levels of the respective attribute and the dependent variable. For each frequency table (i.e., each attribute), a total error is calculated by summing the minima of each level of the attributes. The attribute with the smallest total error is the attribute that is chosen as the best predictor. The rules that are being generated take every level of this predictor and match it with the most frequent class of the dependent variable [17].

Numeric attributes have to be discretized before they can be used by the OneR algorithm. Different discretization methods exist for implementation of the OneR algorithm (package "OneR") used in this paper. Significant further enhancement of the original OneR algorithm is achieved by the discretization methods to optimally align cut points in relation to the dependent variable (function "optbin"). The method "infogain" used here is an entropy-based method taken from information theory, which calculates cut points based on "information gain". The idea is that uncertainty is minimized by making the resulting categories as pure as possible. This method is also the standard method of many decision tree algorithms [18].

Natural generalization of the OneR algorithm is conveyed by decision trees. Though OneR only uses one attribute for its predictions, decision trees are not bounded by this restriction, often resulting in better accuracy but worse interpretability (a trade-off well known in the ML

area) [33]. Further generalization is achieved with random forests, which will not be covered in this paper [35].

There are several different decision tree generation algorithms (e.g., ID3, C4.5, and C5.0), we used CART in the rpart implementation [20]. Unlike linear models, such as Pearson correlation or linear regression, decision trees map non-linear relationships well [36]. Interestingly, the opening example of Breiman's seminal work was a medical example in the area of cardiology [19]. In our population, we used decision trees to create a model predicting mortality or remaining CKD or KRT.

For CART, trees are constructed by repeated splits of subsets of the population into two descendant subsets [19]. With OneR, an attribute can be split into several subsets, but the splits in CART are only binary. For numeric data, cutoff values are determined. The splitting is conducted in a recursive manner, and the same attribute can be used several times on different levels of the resulting tree. Unlike other tree methods and OneR above, the splitting criterion is based not on entropy, but on Gini impurity. In practice, both methods often lead to similar results [23].

An important consideration is the depth of the tree. The deeper a tree, the better it represents the data, but the less interpretable it becomes. An additional problem is overfitting, another well-known problem in the ML literature [37]; a fully grown tree could mean that only one example per leave remains, a result that would render the decision next to useless in practice and would fail to generalize the data (i.e., model the noise in the data). CART prune to an optimal level according to some cost function [38].

We have also performed ANOVA and a standard t-test with the collected parameters. Proteinuria has been commonly observed during SARS-CoV2 infection and is reported in 7 to 63% of cases [25,39]. Proteinuria is mostly reported as unselective due to tubular injury, but in some cases there is a selective proteinuria as an indication of glomerular damage [40]. A direct link between proteinuria and mortality in COVID-19 patients has not been shown, though previous data from critically ill patients due other causes strongly suggest that link [28]. Gross et al. already hypothesized in May that the occurrence of proteinuria could be an early marker of AKI onset or a severe course [41]. Our single center observations included too few patients to transfer this hypothesis to COVID-19, as already mentioned, this is a limitation of our study. This is also a problem for some of the following results, although we were still able to reproduce some findings from previous studies:We found a relationship between higher age and mortality as shown previously in various retrospective studies [5,42,43]. In our study population, the average age of surviving critically ill patients was 60 years, but 69 years among deceased patients. This is also similar to prior results, in which age >65 years was shown to be a risk factor for higher mortality [2,10].

In the ANOVA, we found no significant relationship between selective or non-selective proteinuria and the development of AKI, permanent CKD, or increased mortality or protracted disease progression. Prior data suggest that >40% of the cases are affected with abnormal proteinuria at hospital admission, and 20–40% of the critically ill patients develop AKI [13,39]. In our center, only 9 patients did not experience acute kidney failure (24.3%), and 20 of the affected developed AKI 3 (54.1%). This may be due, among other reasons, to the fact that serum creatinine does not match the baseline creatinine when taken in an already critically ill state. Pei et al. found that 75.4% of 333 patients had abnormal urine dipstick tests or AKI, 50% of them developed AKI 3. Among 35 patients who developed AKI in Guangchang Pei´s work, 45.7% experienced complete recovery of kidney function [26]. We were able to reproduce these results in our patient cohort.

Nine patients died (24.3%), all of them experienced AKI 3 with a need for KRT. In this group, six patients had no or only low grade CKD, and three were admitted with CKD 3b or 4.

Other data reported similar mortality rates among ICU patients [44], especially for patients requiring mechanical ventilation [45]. One meta-analysis showed that the presence of AKI is associated with 13-fold increased risk of mortality, whereas the incidence of AKI is up to 20% in critically ill patients. Higher age, diabetes, hypertension, and baseline serum creatinine levels were associated with increased AKI incidence [46].

Several studies have reported higher BMI as a significant risk factor [47,48]. A meta-analysis by Hussain et al. demonstrated significantly higher mortality in patients with BMI >25 kg/m$^2$, and obesity (BMI >30 kg/m$^2$) as a significant factor for critical illness during COVID-19 [49]. The BMI among deceased patients was 33 kg/m$^2$ in our study population, but it was 28 kg/m$^2$ among surviving patients; nevertheless, this was not significant in our cohort.

From a meta-analysis of a multinational database [50], the incidence of AKI in mechanically ventilated patients was reported to be 22%, slightly higher than among general inpatients [51]. We found an incidence of AKI 1–3 of 75.7%; in 24 of the 37 cases it was >AKI 2 (64.9%). A Chinese meta-analysis reported the incidence of AKI in hospitalized Chinese adults was up to 50% for those in the ICU, and the presence of AKI was associated with a higher severity of infection [52]. Of course, with the small number of cases it is difficult to derive definitive statements. The high incidence of AKI at our center may also be due to patient selection as a center for ECMO therapy [53,54].

Prior data reported that SARS-CoV-2 uses Angiotensin converting enzyme 2 (ACE-2) to enter target cells, which is expressed in lung, liver, oesophagus, gastrointestinal tract, kidney and the cardiovascular system [55–57]. Uncontrolled release of cytokines and other proinflammatory substances are also responsible for either AKI or acute cardiac disorders [58]. In case proteinuria emerges as an early biomarker for AKI in COVID-19 in larger studies, another very interesting approach for future investigation would be to see if potential cardiac deterioration due to COVID-19 can also be detected early this way.

## Conclusion

ML methods are traditionally reserved for large data sets, at least for widespread application. The medical field in particular, where research is mostly based on traditional statistical methods, can provide these large data sets. In the current situation, structured cooperation by all countries is needed, which is reoriented daily to the current and permanently growing knowledge about COVID-19. The methods shown in this paper can be enhanced further by generalization of tree-based methods, especially random forests, which are ensembles of decision trees known for their better accuracy, but unfortunately losing some of the comprehensibility of simpler tree-based methods [35]. Further research is warranted to address this issue.

## Author Contributions

**Conceptualization:** Holger K. von Jouanne-Diedrich.

**Data curation:** Anna Laura Herzog, Holger K. von Jouanne-Diedrich.

**Formal analysis:** Holger K. von Jouanne-Diedrich.

**Investigation:** Anna Laura Herzog, Holger K. von Jouanne-Diedrich.

**Methodology:** Anna Laura Herzog, Holger K. von Jouanne-Diedrich, Patrick Meybohm.

**Project administration:** Anna Laura Herzog, Holger K. von Jouanne-Diedrich.

**Resources:** Anna Laura Herzog, Dirk Weismann, Tobias Schlesinger, Jan Stumpner.

**Software:** Holger K. von Jouanne-Diedrich.

**Supervision:** Holger K. von Jouanne-Diedrich, Christoph Wanner, Patrick Meybohm.

**Visualization:** Anna Laura Herzog.

**Writing – original draft:** Anna Laura Herzog.

**Writing – review & editing:** Holger K. von Jouanne-Diedrich, Christoph Wanner, Patrick Meybohm.

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
