## [Decision Letter · Decision Letter 0]

15 Apr 2021

PONE-D-21-08660

COVID-19 and the kidney: A retrospective analysis of 37 critically ill patients using machine learning

PLOS ONE

Dear Dr. Herzog,

Thank you for submitting your manuscript to PLOS ONE. After careful consideration, we feel that it has merit but does not fully meet PLOS ONE’s publication criteria as it currently stands. Therefore, we invite you to submit a revised version of the manuscript that addresses the points raised during the review process.

We look forward to receiving your revised manuscript.

Kind regards,

Prof. Raffaele Serra, M.D., Ph.D

Academic Editor

PLOS ONE

Journal Requirements:

When possible, we recommend authors deposit restricted data to a repository that allows for controlled data access. If this is not possible, directing data requests to a non-author institutional point of contact, such as a data access or ethics committee, helps guarantee long term stability and availability of data. Providing interested researchers with a durable point of contact ensures data will be accessible even if an author changes email addresses, institutions, or becomes unavailable to answer requests.

As such, we ask you to upload your R code as a supplemental file, or provide a URL to a data repository where the code is hosted.

Thank you for stating the following in the Competing Interests section:

Anna Laura Herzog, Holger K. von Jouanne-Diedrich, Dirk Weismann, Tobias Schlesinger, Patrick Meybohm, and Jan Stumpner have nothing to disclose. Christoph Wanner received honoraria for steering committee membership and lecturing outside the present work from AstraZeneca, Bayer, Boehringer-Ingelheim, Eli Lilly, Mundipharma, and MSD.

We note that you have indicated that data from this study are available upon request. PLOS only allows data to be available upon request if there are legal or ethical restrictions on sharing data publicly. For information on unacceptable data access restrictions, please see http://journals.plos.org/plosone/s/data-availability#loc-unacceptable-data-access-restrictions.

4a) If there are ethical or legal restrictions on sharing a de-identified data set, please explain them in detail (e.g., data contain potentially identifying or sensitive patient information) and who has imposed them (e.g., an ethics committee). Please also provide contact information for a data access committee, ethics committee, or other institutional body to which data requests may be sent.

4b) If there are no restrictions, please upload the minimal anonymized data set necessary to replicate your study findings as either Supporting Information files or to a stable, public repository and provide us with the relevant URLs, DOIs, or accession numbers. Please see http://www.bmj.com/content/340/bmj.c181.long for guidelines on how to de-identify and prepare clinical data for publication. For a list of acceptable repositories, please see http://journals.plos.org/plosone/s/data-availability#loc-recommended-repositories.

Additional Editor Comments:

The manuscript is potentially interesting. Provided the authors are willing to revise the manuscript according to reviewer's suggestions, it will be accepted.

Reviewers' comments:

Reviewer's Responses to Questions

**Comments to the Author**

1. Is the manuscript technically sound, and do the data support the conclusions?

Reviewer #1: Yes

2. Has the statistical analysis been performed appropriately and rigorously? 

Reviewer #1: Yes

3. Have the authors made all data underlying the findings in their manuscript fully available?

Reviewer #1: Yes

4. Is the manuscript presented in an intelligible fashion and written in standard English?

Reviewer #1: No

5. Review Comments to the Author

Reviewer #1: Herzog and Colleagues, with the present Research Article, addressed a very important research question namely whether proteinuria forecasts major renal endpoints (CKD, kidney failure and mortality) in critically ill patients with COVID-19. In my opinion the idea is of interest since controversial results have been reported about this topic showing in particular that overall CKD (but not the exact kidney measure) is a risk factor for mortality. Moreover, the presence of proteinuria has been recognized as a cardiovascular risk factor and data about its role in COVID-19 patients are thus expected. Notwithstanding, I have several concerns to share with Authors:

- Sample size is small, reducing the generalizability of study results: please mention this point among the limitations of the manuscript

- Curiously, I saw that Authors reported in the Results a copy of the syntax used to run the OneR algorithm. This is unusual in the context of an original research. I suggest to delete that part

- Within the machine learning techniques, there are interesting approach which are useful in the case of small sample size and a large amount of data. These include the penalized regression such as the Lasso regression or the Ridge regression. Did you consider using some of them?

- The discussion section could be implemented by generating more hypotheses on the link between kidney measured and cardiorenal risk.To this aim, please read and if possible cite the manuscript “doi: 10.3390/jcm9082506” and “doi: 10.2217/bmm-2020-0201”.

6. PLOS authors have the option to publish the peer review history of their article (what does this mean?). If published, this will include your full peer review and any attached files.

Reviewer #1: **Yes: **Michele Provenzano

---

## [Author Response · Author response to Decision Letter 0]

27 Apr 2021

Dear Prof. Dr. Provenzano

We thank you for the critical evaluation of our work. We are glad that our manuscript was well received (“technically sound, data support the conclusions, very important research question”) and would therefore like to reply to the issues raised by drawing the attention to following items:

Is the manuscript presented in an intelligible fashion and written in standard English? “No”

We apologize that our manuscript does not meet the required language standards. The manuscript was under professional editing, we will ask for a revision from the responsible editor once again

Ad 1.: Sample size is small, reducing the generalizability of study results: please mention this point among the limitations of the manuscript

Thank you for this valuable suggestion. We are absolutely aware that our case counts are too small for a valid statement. Our intention was to provide an overview of the methodology, its application and sources of error. Regarding the predictive power of proteinuria, our results are in line with previous reports despite the small number of patients, although even in much larger studies the exact correlation is not clear. To explain different ML methods, the small sample size was also sufficient, demonstrating sources of error such as overfitting in a simplified way. We have mentioned this again in the discussion.

Ad 2. “Curiously, I saw that Authors reported in the results a copy of the syntax used to run the OneR algorithm. This is unusual in the context of an original research. I suggest to delete that part”

Thank you for this comment. We have tried to explain ML methods in a simple way. The OneR package is an important tool for our approach. We have shown the part of the syntax to give an unfamiliar user an insight into the quite intuitive output of the program. We realize that this is unusual and have corrected that.

Ad 3.: “Within the machine learning techniques, there are interesting approach which are useful in the case of small sample size and a large amount of data. These include the penalized regression such as the Lasso regression or the Ridge regression. Did you consider using some of them?”

Thank you for this important remark. The paper is intentionally focused on machine learning methods, which closes a gap in the current literature. We considered regularization methods for linear regression (e.g. Lasso and Ridge regression), however, this more traditional approach would have gone beyond the scope of this work. A second more technical reason is that the target variable of Lasso and Ridge Regression is numerical whereas our target variables are categorical. Although Lasso and Ridge regression models can be interpreted to some degree they also suffer from some issues. Our approaches (i.e. OneR and decision trees) can be understood very easily and have a clear medical interpretation as we tried to show in the paper.

Ad 4.: “The discussion section could be implemented by generating more hypotheses on the link between kidney measured and cardio-renal risk. To this aim, please read and if possible cite the manuscript “doi: 10.3390/jcm9082506” and “doi: 10.2217/bmm-2020-0201”.

Thank you for this very interesting approach. In fact, we have recorded several cardiac parameters. We could also show that severe AKI in combination with cardiac involvement increases the risk of death. A very interesting approach for our future investigation would be to see if potential cardiac deterioration due to COVID-19 can also be detected early this way, we will follow up on this.

We have largely implemented your valuable suggestions on methods and discussion, added more contemporary literature, structured the main text to make citations more clearly and explained or removed individual sentences. We have changed the manuscript and thank the reviewer for pointing out these important issues.

Kind regards, 

Anna Laura Herzog

On behalf of the authors

---

## [Editor Report · Decision Letter 1]

6 May 2021

COVID-19 and the kidney: A retrospective analysis of 37 critically ill patients using machine learning

PONE-D-21-08660R1

Dear Dr. Herzog,

We’re pleased to inform you that your manuscript has been judged scientifically suitable for publication and will be formally accepted for publication once it meets all outstanding technical requirements.

Kind regards,

Prof. Raffaele Serra, M.D., Ph.D

Academic Editor

PLOS ONE

Additional Editor Comments (optional):

amended manuscript is acceptable
---

## [Editor Report · Acceptance letter]

10 May 2021

PONE-D-21-08660R1 

COVID-19 and the kidney: A retrospective analysis of 37 critically ill patients using machine learning 

Dear Dr. Herzog:

I'm pleased to inform you that your manuscript has been deemed suitable for publication in PLOS ONE. Congratulations! Your manuscript is now with our production department. 

Kind regards, 

on behalf of

Prof. Raffaele Serra 

Academic Editor

PLOS ONE